# Cardiovascular risk factors and its patterns of change between 4 and 8 years of age in the INMA-Asturias cohort

Rocío Fernández-Iglesias[1,2,3], Ana Fernández-Somoano[1,2,3]*, Cristina Rodríguez-Dehli[3,4], Rafael Venta-Obaya[5,6], Isolina Riaño-Galán[1,3,7°], Adonina Tardón[1,2,3°]

**1** Spanish Consortium for Research on Epidemiology and Public Health (CIBERESP), Madrid, Spain, **2** Unit of Molecular Cancer Epidemiology, University Institute of Oncology of the Principality of Asturias (IUOPA), Department of Medicine, University of Oviedo, Oviedo, Asturias, Spain, **3** Instituto de Investigación Sanitaria del Principado de Asturias (ISPA), Oviedo, Asturias, Spain, **4** Servicio de Pediatría, Hospital San Agustín, Avilés, Asturias, Spain, **5** Servicio de Bioquímica, Hospital San Agustín, Avilés, Asturias, Spain, **6** Departamento de Bioquímica y Biología Molecular, University of Oviedo, Oviedo, Asturias, Spain, **7** Servicio de Pediatría, Endocrinología pediátrica, HUCA, Oviedo, Asturias, Spain

☯ These authors contributed equally to this work.
* fernandezsana@uniovi.es

☑ OPEN ACCESS

**Data Availability Statement:** All relevant data are within the paper and its Supporting information files.

## Abstract

### Aim

This study aimed to investigate whether there are subgroups of children with different clusters of cardiovascular disease (CVD) risk factors at 4 and 8 years of age, and their patterns of change between these two time points.

### Methods

The analysis was conducted in 332 children who participated in the INMA-Asturias cohort (Spain) at 4 and at 8 years of age. The CVD risk factors were central obesity, dyslipidaemia, hyperglycaemia, and hypertension. Latent transition analysis was used to identify the different clusters and their probabilities of change.

### Results

At 4 years, three subgroups were identified: no disorders (prevalence of 55.9%); some disorders (21.2%), and central obesity (22.9%). Three distinct subgroups were identified at 8 years: no disorders (59.8%); hypertension (17.9%), and central obesity (22.3%). Central obesity at 4 years tends to appear simultaneously with dyslipidaemia, while at 8 years it tends to appear simultaneously with dyslipidaemia and/or hypertension. Children aged 4 years with no disorders had a 93.7% probability of remaining in the same status at 8 years of age. Children aged 4 who had some disorders had a 67.7% of probability of having only hypertension and a 32.3% of probability of having central obesity. Children aged 4 in the central obesity subgroup had a 32.4% of probability of having no disorders at 8 years of age, while 67.6% still had central obesity.

**Funding:** This study was supported by grants from CIBERESP (PhD-employment-contract), ISCIII: PI04/2018, PI09/02311, PI13/02429, PI18/00909 co-funded by FEDER, "A way to make Europe"/ "Investing in your future", Fundación Cajastur and Universidad de Oviedo. The funders had no role in study design, data collection and analysis, decision to publish, or preparation of the manuscript.

**Competing interests:** The authors have declared that no competing interests exist.

## Conclusions

These exploratory findings suggest that children who do not present any disorder at 4 years of age tend to remain in that state at 8 years of age. And also that central obesity may play a major role in the development of other disorders, as the number of disorders with which it concomitantly occurs increases between the ages of 4 and 8 years.

## Introduction

Cardiovascular disease (CVD) is the leading cause of death and chronic disability throughout the world, almost duplicating the cancer mortality rate in 2017, and it increased by nearly 50% from 1990 to 2019 [1, 2]. Disorders, such as hypertension, dyslipidaemia, central obesity, and hyperglycaemia, are the main risk factors for CVD [2–4], and in the last decade, the prevalence of CVD has been increasing [5]. Moreover, these factors tend to appear simultaneously more frequently than would be expected randomly, and this increases the risk of type 2 diabetes, atherosclerosis, and other CVDs in adults. The clustering of at least three of these CVD risk factors has been defined as a complex disorder called metabolic syndrome (MetS) [6].

The accumulation of CVD risk factors has also been observed in children [7, 8]. However, MetS in childhood is controversial for several reasons. One reason is that there is no consensus on the definition of MetS. Although the same four components (hypertension, dyslipidaemia, central obesity, and hyperglycaemia) are usually considered, the cut-off points used to discriminate normal from abnormal values of this components vary between definitions. This causes difficulty in comparing the prevalence of MetS (range: 0.3%–26.4%) among studies worldwide [8–12]. Additionally, the literature on childhood MetS is scarce and shows inconsistent results on whether the joint presence of components implies a higher risk of disease in adulthood than the simple presence of individual components [13–15]. Therefore, more large-scale longitudinal studies are required to validate a causal relationship. Consequently, most studies conducted to date focussed on all the MetS components separately as CVD risk factors and all of their possible aggregation patterns [10, 16]. Especially at an early age, as the International Diabetes Federation suggests that MetS should not be diagnosed younger than 10 years [17].

Furthermore, the temporal stability of these components and their clustering in children are unclear. The phenomenon of longitudinal stability of a variable is known as tracking [18]. Several studies have examined the tracking of clusters of CVD risk factors, but most focussed on adolescence and the transition of patterns to young adulthood or adulthood [19–25]. To date, there have not been many studies that evaluated the short-term tracking of CVD risk factor clustering in children [26]. One of the main difficulties in assessing the stability of these factors over time is the large number of different observable patterns.

Latent transition analysis (LTA) is a statistical method that allows to identify groups of subjects who are distinguished from each other according to the response to observed categorical variables and estimates the probabilities of transition between groups over time [27–29]. This technique is relatively new in the field of epidemiology [27] but is useful for representing the variety and complexity of the relations between MetS components and their changes over time, taking into account within-person variability [26].

There are future implications of the early onset of CVD risk factors and their clustering [8, 9], and there is uncertainty regarding its pathogenic mechanism [10, 16]. Therefore, identifying the different CVD risk factor clusters in children and their patterns of change over time is important. This study aimed to evaluate the following using LTA: (i) the presence of subgroups

of children with different clustering patterns of hypertension, dyslipidaemia, central obesity, and hyperglycaemia at 4 and 8 years of age, (ii) the prevalence of each subgroup, and (iii) the probability of changing between groups between 4 and 8 years of age.

## Materials and methods

### Study population

The population considered in this study was the Infancia y Medio Ambiente (INMA [Environment and Childhood]) Asturias cohort (northern Spain), which has been described in previous studies [30–32]. In brief, pregnant women in their first trimester of pregnancy were recruited between May 2004 and June 2007. The inclusion criteria were as follows: age $\geq$ 16 years, singleton pregnancy, no assisted conception, delivery scheduled at the San Agustón Hospital (Avilés, Spain), and no communication handicap. After recruitment, data were collected in several phases of follow-up as follows: in the first and third trimesters of pregnancy, at birth, and when children were 18 months, 4 years, 8 years, and 11 years of age. This study analysed data from follow-ups at 4 (T0) and 8 years (T1) of age.

The initial sample was composed of 494 eligible women who agreed to participate. A total of 453 children were followed up at 4 years and 416 children were followed up at 8 years of age. Finally, all 332 children who had available data for each variable used to determine the CVD risk factors (waist circumference [WC], systolic blood pressure [SBP], diastolic blood pressure [DBP], blood glucose, triglycerides [TGs], and high-density lipoprotein cholesterol [HDL-C]) at 4 or at 8 years of age were included in the study (Fig 1).

The study protocol was approved by the Asturias Regional Ethics Committee, and written informed consent was obtained from every participating woman and, in such case, her partner. The research conformed to the principles of the Declaration of Helsinki.

### Cardiovascular disease risk factor definitions

CVD risk factors considered in this study were central obesity, hyperglycaemia, dyslipidaemia, and hypertension. To determine whether a child had these disorders, we considered the following variables. WC was a marker for central obesity, blood glucose concentrations were a marker for hyperglycaemia, TG and HDL-C concentrations were markers for dyslipidaemia, and SBP and DBP were markers for hypertension [33, 34]. Predefined cut-off points were used to distinguish between normal or abnormal levels of these variables.

In this study, reference cut-off points provided by the IDEFICS study were applied. These reference values were derived from a large population-based sample of healthy children from a heterogeneous European population (16,228 children from Sweden, Germany, Hungary, Italy, Cyprus, Spain, Belgium and Estonia). This previous study provided age- and sex-specific cut-off points for each variable (also a height-specific cut-off for blood pressure) using two different levels. One cut-off indicated children who required close observation, called the monitoring level (values exceeding the 90th percentile in their sample). The other cut-off indicated children who required an intervention to ameliorate their risk profile, called the action level (values exceeding the 95th percentile in their sample).

In the present analysis, the cut-off points of the monitoring levels were considered as follows. Children were classified as having central obesity if they had a WC above the age- and sex-specific monitoring IDEFICS cut-off point [35]. Children were classified as having hyperglycaemia if they had blood glucose concentrations above the age- and sex-specific monitoring IDEFICS cut-off point [36]. Children were classified as having dyslipidaemia if they had TG concentrations above the age- and sex-specific monitoring IDEFICS cut-off point or HDL-C concentrations below the age- and sex-specific monitoring IDEFICS cut-off point [37].

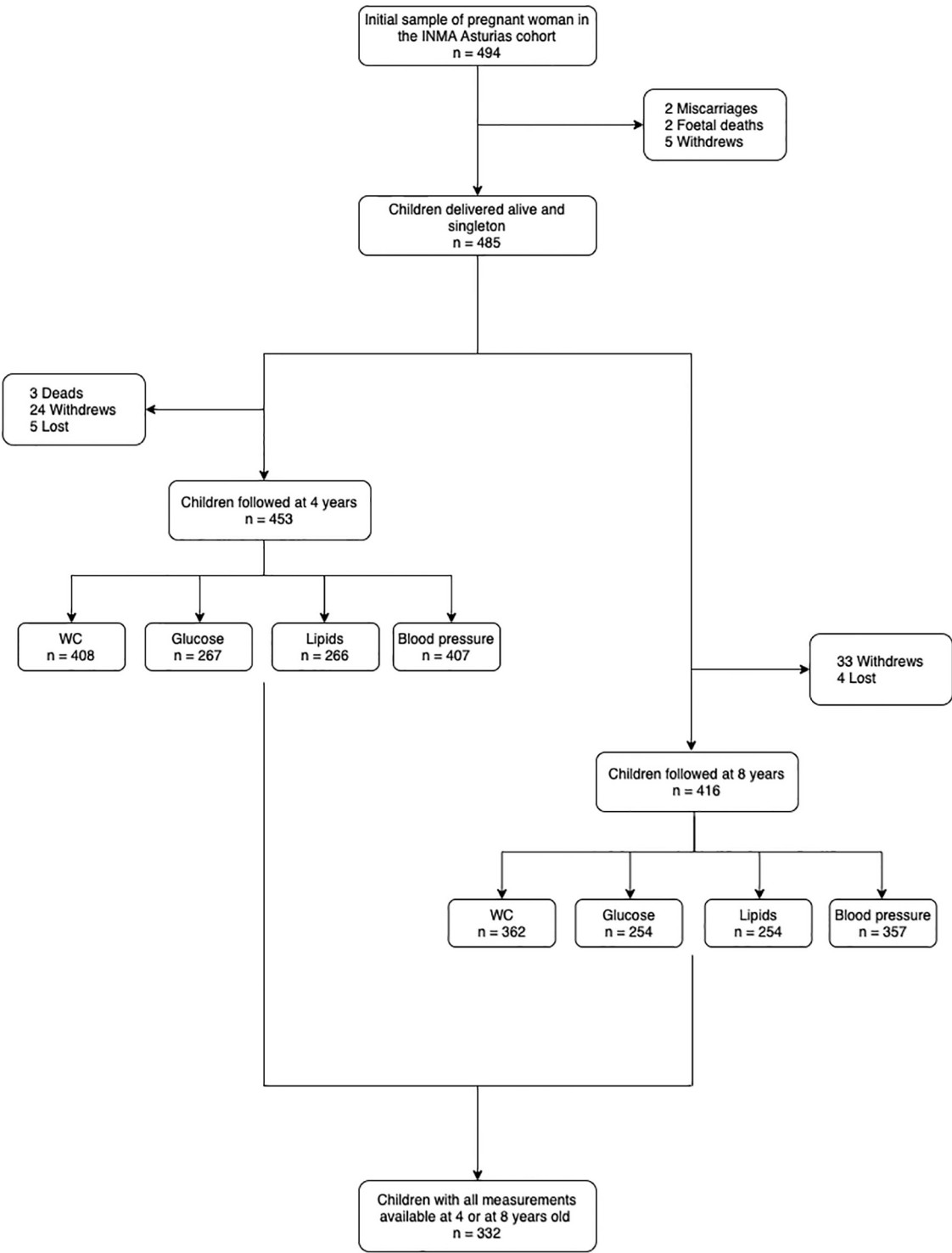

**Fig 1. Flowchart of the study population.** WC, waist circumference.

Children were classified as having hypertension if they had SBP or DBP above the age-, sex-, and height-specific monitoring IDEFICS cut-off point [38].

## Cardiovascular disease risk factor measurement

WC was measured using a non-stretch nylon tape measure, with the subject in a standing position, at the midpoint between the caudal edge of the last rib and the iliac crest at the level of the umbilicus and with the tape in a horizontal plane. SBP and DBP were obtained, after 5 min rest, with an automated oscillometric system (OMRON®) with the patient in a seated position and the right arm at rest at the level of the heart, measured at the level of the right wrist. Between 2 and 3 measurements were taken and the mean was calculated. All somatometric and blood pressure determinations were performed by experienced and trained healthcare staff. Nonfasting blood samples were collected by laboratory nurses at the Hospital San Agustín de Avilés and determined by laboratory staff using a Roche analyser (Modular Analytics Serum Work Area, Mannheim, Germany).

## Statistical analysis

The characteristics of the study population are shown using descriptive statistics. Continuous variables are shown by median and interquartile range, while categorical variables are shown by absolute and relative frequencies.

As the result of the definitions indicated in the 'Cardiovascular disease risk factor definitions' section, four binary categorical variables were obtained that allowed us to distinguish children who had normal or abnormal (monitoring) levels in relation to the four disorders. LTA was used to identify groups of children with different aggregation patterns of CVD risk factors. Each identified group is interpreted as a 'latent status', that represents the different response patterns in the data to the observed variables [29], which are in this case the four CVD risk factor markers considered. LTA was also used to estimate the prevalence of each latent status and the probabilities of change from one latent status to anotherbetween these two time points (S1 Figure in S1 File). In this process, three types of parameters were estimated as follows. Class membership probabilities, which indicate the proportion of the population expected to be classified in a particular latent status. Item-response probabilities, which indicate the probability of being in a particular category of an observed variable (i.e., in the normal or monitoring level category of any of the disorders), conditional to the latent class membership. They also provide the basis for interpretation and labelling the latent status. And finally, transition probabilities, which indicate the probability of membership in one latent status at T1 given the membership at T0. The name given or assigned to the latent status is based on the researchers interpretation of the item-response probabilities. LTA model specification, model estimation, and finally, model selection and interpretation were conducted as reported by Collins and Lanza [29], with an adaptation of the procedure proposed by Ryoo et al. [39]. Further details of the steps followed during this procedure are given in S1 File.

LTA models allow for missing data on the measured outcomes using the full-information maximum likelihood method. In this method, the data for children who have information at only one time point do not contribute to the estimation of the transition parameters. However, the data do contribute to estimating the time-specific parameters, and thus help to produce better results, allowing an increase in the sample size and thus the statistical power [29, 40]. Therefore, only children without any of the measurement components at any time point were eliminated from the study. Consequently, children were included in the study if they met the following criteria: they had a measurement of each of the CVD risk factors considered in at least one of the time points (i.e., those who have measure of WC at T0 or T1, and measure of

blood glucose at T0 or T1, and measure of TG at T0 or T1, and measure of HDL-C at T0 or T1, and measure of SBP at T0 or T1, and measure of DBP measurement at T0 or T1). This resulted in a final study sample of 332 children, which allowed us to reach a sample size of > 300 (for lower sample sizes, the use of LTA models is not recommended [41]). To estimate transition probabilities, we used children who had measurements of all variables at the two time points (n = 154).

Statistical analyses were performed using the R software (version 4.0.5) [42] and the statistical modelling programme Mplus (version 7) [43], which is specially designed for latent variable models. LTA was carried out using Mplus and the R package MplusAutomation [44].

## Results

Descriptive statistics of sex, age, cardiometabolic variables, and risk factors of the study population are shown in Table 1. The risk factors with the highest prevalence at T0 at the monitoring level were blood lipids and WC, with a prevalence of 33.7% and 27.1%, respectively. The risk factors with the highest prevalence at T1 were blood pressure and lipids, with a prevalence of 35.8% and 25.0%, respectively.

The distribution of the number of risk factors at the monitoring level that a child presented at the same time (Fig 2) at T0 was as follows. A total of 29.0% of the children had no disorder at this level, 41.6% had one, 21.2% had two, 8.0% had three, and 0.4% had all four. At T1, a higher percentage of children presented with none or one disorder at that level (36.9% and 42.1% respectively), while a lower percentage of children had two or three disorders at that

**Table 1. Descriptive statistics of sex, age and cardiometabolic parameters at 4 and 8 years of age.**

| | N = 327 | T0 (4 years) | N = 300 | T1 (8 years) |
|---|---|---|---|---|
| Sex, % | | | | |
| Male | 176 | (53.8) | 162 | (54.0) |
| Female | 151 | (46.2) | 138 | (46.0) |
| Age, median (IQR[a]) | 327 | 4.4 (4.3; 4.5) | 300 | 8.3 (8.1; 8.4) |
| Waist circumference (cm), median (IQR) | 325 | 53.5 (50.5; 56.0) | 300 | 63.2 (58.5; 69.0) |
| Glucose (mg/dL), median (IQR) | 266 | 86.0 (81.2; 91.0) | 253 | 86.0 (82.0; 90.0) |
| Systolic blood pressure (mmHg), median (IQR) | 324 | 99.0 (90.0; 105.0) | 296 | 107.0 (100.0; 114.0) |
| Diastolic blood pressure (mmHg), median (IQR) | 324 | 60.0 (54.0; 66.0) | 296 | 67.0 (60.8; 72.0) |
| Triglycerides (mg/dL), median (IQR) | 265 | 71.0 (55.0; 94.0) | 253 | 64.0 (47.0; 87.0) |
| HDL-C (mg/dL), median (IQR) | 265 | 57.0 (48.0; 63.0) | 251 | 68.0 (60.0; 80.0) |
| Waist circumference level, % | | | | |
| Normal | 237 | (72.9) | 230 | (76.7) |
| Monitorization | 88 | (27.1) | 70 | (23.3) |
| Glucose level, % | | | | |
| Normal | 210 | (79.2) | 237 | (93.7) |
| Monitorization | 55 | (20.8) | 16 | (6.3) |
| Blood pressure level, % | | | | |
| Normal | 238 | (73.5) | 190 | (64.2) |
| Monitorization | 86 | (26.5) | 106 | (35.8) |
| Lipids level, % | | | | |
| Normal | 175 | (66.3) | 189 | (75.0) |
| Monitorization | 89 | (33.7) | 63 | (25.0) |

[a]IQR, interquartile range.

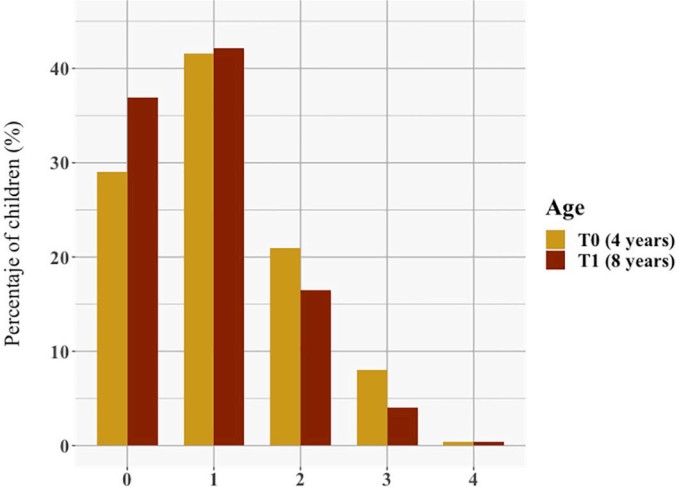

**Fig 2. Distribution of the number CVD of risk factors at the monitoring level in the study population.**

level (16.4% and 4.0% respectively). The percentage of children with the four risk factors at that level was the same at both time points. Consequently, 71.2% of children had at least one disorder at the monitoring level at 4 years and 62.9% of children did so at 8 years.

S1 Table in S1 File also contains the distribution of the number of risk factors at the monitoring level at the same time, disaggregated by each risk factor. It shows that at 4 years WC and blood lipids are the variable that appeared at the monitoring level most frequently in combination with others. And that at 8 years WC continued to be the variable that appeared at the monitoring level most frequently in combination with others, followed by blood pressure.

The results of the LTA are described below. On the basis of the criteria indicated in the S1 File, the three latent status model was selected. Details about the information on the relative model fit for selecting the number of latent statuses is shown in Supplementary Table S2 in S1 File.

Table 2 shows the item-response probabilities of being at a normal level for each disorder, in the three latent statuses detected at each time point. It also shows the estimated prevalence of each latent status. On the basis of these item-response probabilities, the latent status detected by LTA was labelled as follows at T0. Children in latent status one were characterised as having a high probability of being in the normal range for all of the disorders (70.9%–100%), so it was labelled as '*no disorders*'. Children in latent status two had the probability of being at normal levels for all of the disorders (48.9 and 79.5%). But all confidence intervals contained the probability of 50%, and therefore, individuals in this group could present with one or more disorders, but were not characterised by any specific one. Consequently, this status was labelled as '*some disorders*'. In latent status three, which was labelled as '*central obesity*', children had a zero probability of having a normal WC, a high probability of having a normal blood pressure (81.3%), and a high probability of having normal glucose concentrations (76.9%), while they were half as likely to have normal lipid concentrations. The second latent status (some disorders) had a poor homogeneity because all of the confidence intervals contained a 50% probability. That means that no clearly characteristic pattern can be identified in this latent status and that is difficult to interpret it in a meaningful way. The latent status with the highest prevalence at T0 was no disorders (55.9%), followed by central obesity (22.9%) and some disorders (21.2%).

**Table 2. Item-response probabilities and confidence intervals of the three latent status model selected, and the prevalence of the latent statuses at each time point.**

| | T0 (4 years) | | |
|---|---|---|---|
| | Latent status 1: No disorders | Latent status 2: Some disorders | Latent status 3: Central obesity |
| | N = 186 (55.9%) | N = 70 (21.2%) | N = 76 (22.9%) |
| Waist circumference normal level (%) | **100.0 (100.0; 100.0)** | 79.5 (47.9; 100.0) | **0.0 (0.0; 0.0)** |
| Glucose normal level (%) | **87.2 (79.8; 94.6)** | 60.5 (37.1; 83.9) | **76.9 (65.2; 88.6)** |
| Blood pressure normal level (%) | **79.4 (72.2; 85.5)** | 48.9 (10.5; 87.4) | **81.3 (66.7; 93.5)** |
| Lipids normal level (%) | **70.9 (60.0; 81.9)** | 69.6 (39.9; 99.2) | 53.0 (38.5; 67.5) |
| | T1 (8 years) | | |
| | Latent status 1: No disorders | Latent status 2: Hypertension | Latent status 3: Central obesity |
| | N = 199 (59.8%) | N = 59 (17.9%) | N = 74 (22.3%) |
| Waist circumference normal level (%) | **92.5 (85.7; 99.2)** | **88.0 (70.6; 100.0)** | 24.0 (0.0; 51.7) |
| Glucose normal level (%) | **93.7 (89.1; 98.2)** | **92.7 (83.0; 100.0)** | **94.5 (86.5; 100.0)** |
| Blood pressure normal level (%) | **87.3 (71.2; 100.0)** | **0.0 (0.0; 0.0)** | 53.5 (34.2; 72.7) |
| Lipids normal level (%) | **75.5 (67.6; 83.5)** | **99.1 (85.6; 100.0)** | 54.2 (36.6; 71.8) |

The item-response probabilities shown correspond to the 'normal' category of each disorder. Item-response probabilities corresponding to the 'monitoring' category are the complements of those corresponding to the 'normal' category; therefore, they are not reported here. Statistically significant estimates are shown in bold. Data are expressed as a percentage.

At T1, latent status one was characterised by the equivalent of the first latent status at T0 and was also labelled as no disorders. Latent status two at T1 was characterised by a zero probability of having normal blood pressure and a high probability of having normal levels in the rest of disorders, so it was labelled as '*hypertension*'. Children in latent status three were characterised as having a low probability of having a normal WC (24.0%), and therefore, this latent status was labelled as "*central obesity*". The difference between this status and the identically named status at T0 is that, in this case, only glucose concentrations had a high probability of being in the normal range. Therefore, subjects in the status of central obesity at T0 could present central obesity alone or accompanied by dyslipidaemia, while at T1, they could suffer from central obesity alone or accompanied by hypertension and/or dyslipidaemia. The latent status with the highest prevalence at T1 was the no disorders status (59.8%), followed by the central obesity status (22.3%) and the hypertension status (17.9%). WC and blood pressure at T1 were strongly related to the children's latent status because these variables clearly differentiated in which latent status a child would be classified.

Transition probabilities of change from one latent status to another between T0 and T1 are shown in Fig 3. The no disorders status (latent status one) showed high stability, with a 93.7% probability that a child without disorders at T0 would remain with no disorders (latent status one) at T1. Among children with an altered latent status, the subgroup with some disorders at T0 (latent status two) had a 67.7% probability of being in the hypertension status (latent status two) at T1 and a 32.3% of probability of being in the central obesity status (latent status three) at T1. The subgroup of central obesity (latent status three) at T0 had a probability of 67.6% of being in the central obesity status (latent status three) at T1 and a probability of 32.4% of being in the no disorders status (latent status one) at T1.

## Discussion

In the present study, three latent statuses were identified at 4 and at 8 years of age, and they were different between these two periods. At both years of age, there was a predominant pattern defined by the presence of no disorders, which was highly stable. At 4 years of age, we also

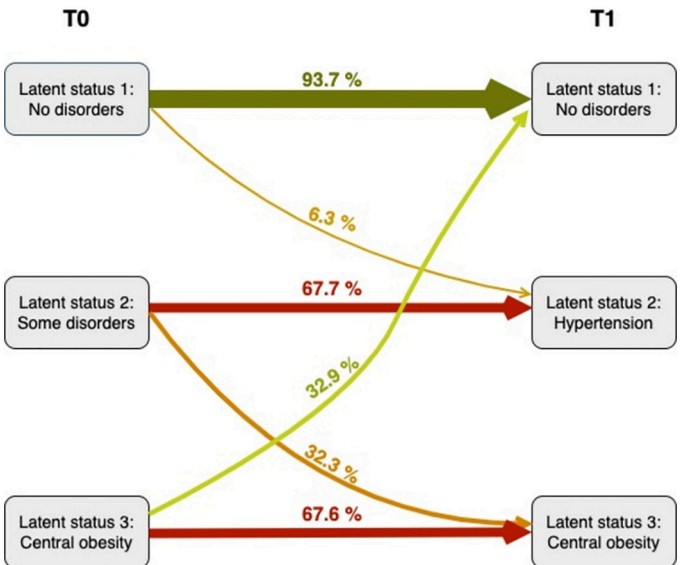

**Fig 3. Probability of transitioning to each latent status at T1 (8 years), conditioned by the latent status at T0 (4 years).** There were 154 subjects who had measurement of all variables performed at the two time points. Data are expressed as a percentage.

identified a subgroup characterised by children who had some disorders and by a subgroup that presented with central obesity and could be accompanied by dyslipidaemia. Over time, children transitioned to one subgroup defined by only hypertension and to another subgroup defined by central obesity, which could be accompanied by dyslipidaemia and/or hypertension (in addition to the no disorder subgroup already mentioned). These results support the recommendation of not defining MetS at paediatric ages [17] because no patterns with aggregations of three or more of the disorders of hypertension, central obesity, hyperglycaemia, and dyslipidaemia were clearly identified.

The lack of a standard criterion at paediatric ages to define cut-off points that discriminate between normal or abnormal levels of the variables used in the present study (WC, glucose concentrations, lipid concentrations, and blood pressure), which enable definition of the above-mentioned disorders, causes difficulty in comparing the prevalence between studies [12]. The reason for this difficulty is the high variability depending on the definition used [7, 9, 45]. In this study, the definition provided by the IDEFICS study was used because, although it is not the most widely used, it is applicable to children aged 2–10.9 years. In the IDEFICS study, cut-off points were calculated from a large and healthy European population that covered southern European countries, such as Spain, in which the sample of this study was based.

Taking into account that the cut-off values used to discriminate between normal and abnormal (monitoring levels) for each of the CVD risk factors derived from IDEFICS study were based on the 90th percentile, we expected that 10% of children would have abnormal levels of these disorders. However, in our sample, we observed that the prevalence ranged from 20.8% to 33.7% at 4 years of age, and from 6.3% to 35.8% at 8 years of age. These prevalences are higher than those observed in the reference population [8] and in other European populations in which the same definitions were applied [26]. An exception was glucose concentrations at 8 years of age in our study, which had a lower prevalence (6.3%). In the reference population on which the cut-off points were calculated those who were overweight or obese were excluded, and it has been observed that the prevalence of some of these disorders is positively associated

with overweight and obesity [7, 8, 14, 45–48]. The fact that the INMA-Asturias cohort is char-
acterised by a high prevalence of overweight and obesity [32] could explain the high prevalence
of these CVD risk factors in monitoring levels compared with the reference population.

Regarding the number of disorders at the monitoring level, 62.9% of the children at 4 years
and 71.2% at 8 years of age presented with at least one component at abnormal levels. These
data are consistent with those reported in other studies, in which approximately two thirds of
the study population had at least one altered disorder [48, 49]. However, because the defini-
tions and cut-off points used are not the same, comparisons should be considered with cau-
tion. Importantly, an improvement in the number of disorders present in the same child was
observed between 4 and 8 years of age because the percentage of children who did not present
with any CVD increased, that of children who presented with only one CVD risk factor
remained the same, and that of children who presented with more than one CVD risk factor
simultaneously decreased.

Three latent statuses were identified at 4 years, namely no disorders, some disorders, and
central obesity. The no disorders status had a prevalence of 55.9%, and the other two latent sta-
tuses had a prevalence of approximately 20%. The central obesity latent status was clearly
defined by the presence of central obesity, but the probability of having normal lipid concen-
trations was approximately 50%. This finding indicated that central obesity could appear alone
or be accompanied mainly by dyslipidaemia. The latent status labelled as some disorders has a
difficult interpretation due to its poor homogeneity. As all the variables have around 50% of
probability of being in the normal category, there is a high degree of uncertainty about what
is being captured in that group and no specific pattern can be identify. But that also give us
some information. As no other latent status (apart from the central obesity) characterised by a
single disorder was found at 4 years, despite the fact that 41.6% of the children had only one
disorder, the existence of this some disorders status suggests there is no one disorder that pres-
ents a higher prevalence than the others in isolation.

Three latent statuses were identified at 8 years, namely no disorders, hypertension, and
central obesity. The no disorders status was the most prevalent, with almost two thirds (59.1%)
of the children in that group. At this age, two latent statuses were characterised by the single
disorder of hypertension, with a prevalence of 17.9%, and central obesity, with a prevalence
of 22.3%. In the case of central obesity, the confidence intervals of the item-response probabili-
ties of normal blood pressure and normal lipid concentrations (confidence interval: 34.2%,
72.7%; and confidence interval: 36.6%, 71.8%, respectively) indicated that central obesity could
be partially accompanied by hypertension and dyslipidaemia.

The latent statuses identified at 4 and 8 years of age were distinct (except in the no disorders
status), suggested a lack of stability of these disorders between these ages, either in isolation or
in clusters. The stability of CVD risk factors associated with MetS at older ages has been dis-
cussed in several articles, although mostly from childhood and adolescence to young adult-
hood and adulthood, instead of between different ages within childhood [13, 19–26]. While
the results of these studies are not consistent, most of them are consistent with the lack of sta-
bility that we observed [13, 19–23, 26]. The main difference between the time points is that, at
4 years of age, there was no pattern characterised by an isolated disorder, and central obesity
could be accompanied by dyslipidaemia. However, at 8 years of age, the detected patterns
changed, a latent status composed only of hypertension appeared to be relevant and the latent
status of central obesity could be accompanied by dyslipidaemia and/or hypertension. There-
fore, the number of disorders that appear at the same time as central obesity increases between
4 and 8 years of age. These findings support the hypothesis that central obesity is a CVD risk
factor that triggers other disorders, which usually emerge as comorbidities of central obesity
[11, 14, 17, 33]. Regarding to hypertension, the increasing trend of elevated blood pressure

levels in childhood and adolescence in terms of primary hypertension has been mainly related to the also increasing trend of childhood obesity, but also to other factors such as prenatal and postnatal exposures, genetic factors, birth characteristics, dietary habits, or lifestyle factors [50]. Some of them have been shown to be related to primary hypertension independently of obesity, as evidenced for example by Rosner et al. [51] in the case of high sodium intake. Specifically, it has also been found that the Spanish pediatric population exceeds the recommended sodium intake at ages 9 to 12 years regardless of their nutritional status [52], and that it is positive associated with an elevated diastolic blood pressure between 5 and 16 years old, also regardless of their nutritional status [53]. So the latent status characterized only by hypertension is evidenced the group of children with high blood pressure values related to causes independent of obesity, since those who present elevated blood pressure values accompanied by obesity would be classified in the latent status of central obesity.

There are two major similarities between the latent status at 4 and 8 years of age. The no disorders status was observed in slightly more than half of the sample, and the other status, which was defined by central obesity accompanied by other disorders, represented approximately one quarter of the sample at the two time points.

Transition probabilities showed that the no disorders status in children at 4 years of age remained mostly in the same status at 8 years of age, because only a small proportion of children transitioned to other statuses at 8 years (6.3% change to the hypertension status). This observation is in line with other studies, which reported that the group of children who were in a healthy or lower metabolic risk status at the start of the follow-up were more likely to remain in that group [25, 26].

Among the children who suffered from some disorders at 4 years, 67.7% transitioned to have only hypertension at 8 years, while 32.3% transitioned to have central obesity, which could be accompanied by hypertension and/or by dyslipidaemia. The finding that hyperglycae-mia disappeared at 8 years of age, because all of the latent statuses at 8 years had a high proba-bility of normal glucose concentrations, could be due to the fact that, at 4 years, the stress caused in such young children by blood collection may produce an unrealistic elevation in glucose concentrations. Additionally, this disappearance could also have been affected by the importance that parents gave to this warning sign, taking measures to reverse it.

Children with central obesity at 4 years of age maintained central obesity at 8 years of age, with a 67.6% probability. However, 32.4% of these children reversed their status by moving to the no disorders status.

Comparing the results of this study with those in other studies with the same or similar objectives is difficult because of the wide variety of variables considered, cut-off points used, diversity in methodology applied, and follow-up periods of the populations studied. We only found one related study in childhood with the use LTA by Bornhorst et al. [26] who analysed the latent status and its transition at 6, 8, and 12 years of age. When we focussed on the results of this previous study between 6 and 8 years of age, because they are the most comparable with our sample, the latent statuses detected were different regarding their pattern and number. However, this previous study also observed a similar prevalence and a higher stability of the no disorders status. Furthermore, central obesity appeared to be the most likely to be accompa-nied by other disorders, as in our study. The differences detected between the other latent sta-tuses in the Bornhorst et al.'s study and ours may be due to the fact that, in our study, no latent statuses were identified with a prevalence of < 10%. Additionally, the smaller sample size of our study may not have allowed us to distinguish underlying patterns. Moreover, in Bornhorst et al.'s study [26], item-response probabilities were restricted to be equal across time, which meant that the identified latent statuses were forced to be equal at the different time points, unlike in our study.

The present study has several limitations. A larger sample size for this study would have been preferable because LTA is a methodology that uses multiple combinations of categorical variables. There may have been different patterns or combinations of responses with a low or even null frequency, which could have affected the results of the estimates and the width of the confidence intervals. The sample size did not allow us to carry out a stratified analysis by sex, which would have been desirable, because sensitivity analysis due to sex-based differences has been observed in some studies [8, 9, 20]. Additionally, the sample size did not enable the introduction of explanatory variables to estimate their effect on the transition probabilities. With regard to the method of defining the disorders at normal levels, in this study, the variables were dichotomised. This could have resulted in a loss of information. Several authors have suggested that CVD risk factors should be treated as continuous [8, 9, 54, 55]. Another limitation is that, because of the lack of longitudinal studies from childhood to adulthood, there are no cut-off points defined on the basis of a relevant and quantifiable increase in the CVD risk in the future or on the basis of biological evidence [8]. Moreover, in this study, the application of cut-off values from a pre-existing reference population, which was different to the population from which the study sample was drawn, could have overestimated or underestimated the prevalence of the disorders. This possibility suggests the necessity of cut-off points that are ethnicity-specific in each country [56]. Finally, blood samples were not collected after 12 hours of fasting. Recent studies have shown that lipid profiles only minimally change in response to normal food intake in individuals in the general population [57], but they could have a greater effect on glucose measurements. Therefore, the results concerning hyperglycaemia should be treated with caution [58].

There are some strengths of this study. Despite using a different reference population, the chosen population included subjects from Spain and other southern European countries, as well as sex- and age- and even height-specific cut-off points. Therefore, this takes into account the physiological changes in childhood, which is a period in which several modifications that affect cardiometabolic parameters occur [34]. Moreover, this study provides knowledge on the evolution and stability of aggregations of CVD risk factors at the paediatric age through a longitudinal study. There have not been many studies that have conducted this analysis during childhood [19, 25, 26, 59, 60], with most studies from childhood or adolescence to young adulthood or adulthood. Therefore, taking into account the existing gaps in knowledge regarding the interrelations between CVD risk factors and their joint appearance, the information provided by the current study could help to provide further understanding on this topic.

## Conclusions

The cluster patterns of different CVDs risk factors are not maintained between the ages of 4 and 8 years, except for those in children who have no disorders. At 8 years of age, the prevalence of hypertension is high and occurs in isolation. Central obesity is found at these two ages. Therefore, the early detection of central obesity has importance for the correct control of its evolution. In addition, central obesity should play a major role in the prevention of the development of other CVD disorders because it is accompanied by other disorders, and the number of disorders that may accompany it increases from the age of 4 to 8 years. The next steps regarding this issue should be focussed on attempting to understand what underlying factors could explain the changes in the latent status throughout childhood.

## Supporting information

**S1 Data.**
(TXT)

**S1 File.**
(DOCX)

## Acknowledgments

The authors would particularly like to thank all of the participants and the families for their generous participation in the study. The authors are grateful to the medical board, the Departments of Gynaecology and Paediatrics in Hospital San Agustín de Avilés, and the Health Centre of Las Vegas in Corvera de Asturias for their disinterested involvement in the project. We thank Ellen Knapp, PhD, from Edanz (https://edanz.com/ac) for editing a draft of this manuscript.

## Author Contributions

**Conceptualization:** Ana Fernández-Somoano, Isolina Riaño-Galán, Adonina Tardón.

**Data curation:** Rocío Fernández-Iglesias.

**Formal analysis:** Rocío Fernández-Iglesias.

**Funding acquisition:** Adonina Tardón.

**Investigation:** Rocío Fernández-Iglesias, Ana Fernández-Somoano, Isolina Riaño-Galán.

**Methodology:** Rocío Fernández-Iglesias, Ana Fernández-Somoano.

**Project administration:** Adonina Tardón.

**Resources:** Rafael Venta-Obaya.

**Supervision:** Ana Fernández-Somoano, Adonina Tardón.

**Visualization:** Rocío Fernández-Iglesias.

**Writing – original draft:** Rocío Fernández-Iglesias.

**Writing – review & editing:** Rocío Fernández-Iglesias, Ana Fernández-Somoano, Cristina Rodríguez-Dehli, Rafael Venta-Obaya, Isolina Riaño-Galán, Adonina Tardón.

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
