## [Decision Letter · Decision Letter 0]

31 Jan 2023

PONE-D-22-28738Cardiovascular risk factors and its patterns of change between 4 and 8 years of age in the INMA-Asturias CohortPLOS ONE

Dear Dr. Fernández Somoano,

Thank you for submitting your manuscript to PLOS ONE. After careful consideration, we feel that it has merit but does not fully meet PLOS ONE’s publication criteria as it currently stands. Therefore, we invite you to submit a revised version of the manuscript that addresses the points raised during the review process.

 Please revise the manuscript according to the reviewers' comments, I want to underline the following:-Abstract needs restructuring-The results needs many clarifications as suggested Reviewer 2 ==============================

We look forward to receiving your revised manuscript.

Kind regards,

Antoine Fakhry AbdelMassih

Academic Editor

PLOS ONE

Journal Requirements:

Reviewers' comments:

Reviewer's Responses to Questions

**Comments to the Author**

1. Is the manuscript technically sound, and do the data support the conclusions?

Reviewer #1: Yes

Reviewer #2: Partly

2. Has the statistical analysis been performed appropriately and rigorously? 

Reviewer #1: No

Reviewer #2: I Don't Know

3. Have the authors made all data underlying the findings in their manuscript fully available?

Reviewer #1: Yes

Reviewer #2: Yes

4. Is the manuscript presented in an intelligible fashion and written in standard English?

Reviewer #1: Yes

Reviewer #2: Yes

5. Review Comments to the Author

Reviewer #1: Review comments on Manuscript Number: PONE-D-22-28738. Entitled "Cardiovascular risk factors and its patterns of change between 4 and 8 years of age in the INMA-Asturias Cohort"

Overall, the idea of research is very interesting to be studied nowadays and paper is coherently developed. However, there are some comments and suggestions.

Title

- Well structured

Abstract

- It is recommended to write the abstract in organized sections (background, aim, methods, results and conclusion)

- Conclusion is rephrasing of the results. It is recommended to provide your conclusion and comment on the findings

- Keywords: write it in alphabetical order

Introduction

- Well structured

Materials and methods

- Well structured

Statistical analysis

- Well structured. However, too many details are written you may need to concise this section.

Discussion

- Well structured

Reviewer #2: What is meant by: Some disorders?? This expression has no clear definition.

Results section:

Table 2: not clear.

It seems that the data of blood levels rather than OR or probabilities.

For examples:

Latent status 1 with no disorders:

Normal lipid profile of 70, it is probably a blood level rather than % of normal.

If 70% of this group which is supposed to be normal has normal lipid profile.

It is confusing how 30% of the normal group has abnormal lipid profile,

Figure 2: risk factors are not labelled and cannot be discriminated.

Despite the high prevalence of hypertension at T1 in the last illustration of the paper.

It is not logical how it is not linked to obesity.

6. PLOS authors have the option to publish the peer review history of their article (what does this mean?). If published, this will include your full peer review and any attached files.

Reviewer #1: No

Reviewer #2: No

---

## [Author Response · Author response to Decision Letter 0]

22 Feb 2023

We want to thank the Academic Editor for giving us the opportunity to review our manuscript. We would also like to thank the reviewers for their helpful comments and suggestions. The revised version of the manuscript “Cardiovascular risk factors and its patterns of change between 4 and 8 years of age in the INMA-Asturias Cohort” is attached. We have reviewed the manuscript in detail, taking into account the comments of the reviewers. We believe the article has been substantially improved as a result. Our responses to reviewers’ comments are provided on the following lines.

Review Comments to the Author

First, we are very grateful for the reviewer's constructive and valuable comments. We have tried to answer all the questions and suggestions raised and believe that the article has been improved thanks to the reviewers' recommendations.

Reviewer #1: 

Review comments on Manuscript Number: PONE-D-22-28738. Entitled

"Cardiovascular risk factors and its patterns of change between 4 and 8 years of age in the

INMA-Asturias Cohort" Overall, the idea of research is very interesting to be studied nowadays and paper is coherently developed. However, there are some comments and suggestions.

Title

- Well structured

Abstract

- It is recommended to write the abstract in organized sections (background, aim, methods,

results and conclusion).

Response: Thanks for the recommendation. The abstract has been organized by adding the title of each section in the corresponding paragraph. 

- Conclusion is rephrasing of the results. It is recommended to provide your conclusion and

comment on the findings.

Response: Based on the reviewer’s comment we have restructured the results and conclusions parts of the abstract. Both parts were mixed, which was confusing. We have clarified the results part and added the appropriate conclusions.

- Keywords: write it in alphabetical order.

Response: Done according to suggestion. We have written it in alphabetical order. 

Introduction

- Well structured

Materials and methods

- Well structured

Statistical analysis

- Well structured. However, too many details are written you may need to concise this section.

Response: Thank you for your annotation. Statistical analysis section has been abbreviated to be more concise. We have excluded the details regarding to the model specification, estimation, and selection. However, we consider that these details, although not essential, could be useful for someone who wants to replicate this analysis to have a guide or to have all the necessary information to compare their results with those shown in our work. So, we have moved the excluded details to the Supplementary file. 

Consequently, we have move to the Supplementary file also the Results parts in which we present information related to model selection (part of the third paragraph and fourth paragraph). We consider it is more consistent to summarize all the information on the statistical details in the same place. 

Discussion

- Well structured

Reviewer #2: 

What is meant by: Some disorders?? This expression has no clear definition.

Response: Thank you so much for your comment. The names assigned to each latent status are a result of the interpretation that the researchers have made about what that state 'captures' or 'represents', based on the item-response probabilities. Someone else might have made a different interpretation or chosen a different name. 

Children in the latent status two, which is the one we labelled as ‘some disorders’, have around a 50% of probability of being in a normal level for the four cardiovascular risk factors markers considered in the study. This indicates that if a child is in that status, we have a very high degree of uncertainty to whether he or she is in the normal or in the monitoring category of any of the variables of interest. When this occurs, it is said that the model presents a low degree of homogeneity and that this latent status does not provide us with much information. It would be desirable to obtain a model with a high degree of homogeneity, i.e., where the probabilities of each observed variable in each latent status are close to 0 or 1. But when this does not occur, it means that no specific pattern is being identified in that group in relation to what is being studied, which in this case is the presence of normal values or monitoring in markers of cardiovascular risk factors. Taking all this into account, latent state two can be interpreted as the group of children in which no specific pattern of absence or presence of disorders is found. But the fact that no specific pattern is found also provides us with information, since it means that in this group there may be children with some disorders, but no greater prevalence of one or more than one of the disorders than the others is observed. We have therefore decided to label it generically as "some disorders", thus indicating that there are disorders in this group, but none is more representative than the others.

To try to clarify this point we have introduced some changes in the Statistical analysis section, mainly by adding the following sentence: “The name given or assigned to the latent status is based on the researcher’s interpretation of the item-response probabilities”. We have also tried to clarify it in the Results section that the latent statuses were labelled based on the interpretation of the items-response probabilities, and we have adding extra explanation to the sentence regarding the poor homogeneity of the latent status two: “The second latent status (some disorders) had a poor homogeneity because all of the confidence intervals contained a 50% probability. That means that no clearly characteristic pattern can be identified in this latent status and that is difficult to interpret it in a meaningful way.” In the Discussion section we have modified the fifth paragraph explaining more the ‘some disorder’ status. If the reviewer considers that we should add more detail in the document about that latent status, as we have done in this response, there would be no problem in doing so. There would also be no objection to changing the label if the reviewer considers that another label is more correct, based on the item-response probabilities interpretation. 

Results section:

- Table 2: not clear. It seems that the data of blood levels rather than OR or probabilities.

For examples:

Latent status 1 with no disorders:

Normal lipid profile of 70, it is probably a blood level rather than % of normal.

If 70% of this group which is supposed to be normal has normal lipid profile.

It is confusing how 30% of the normal group has abnormal lipid profile,

Response: Thank you so much for your comment. Table 2 shows the probability that a child is in the ‘normal’ category for each variable considered, conditional on each of the three latent status. In LTA theory these probabilities are called “item-response probabilities” and they are defined in the Statistical analysis section of the paper. We have added the symbol of the percentage in this table to clarify what it contains.

Regarding your remark about how confusing these probabilities can be, we completely understand. It could be confusing. The key point here that can be confusing or difficult to understand, is that LTA models classified children in groups, but this classification is probabilistic. In other words, the model does not “know” in which latent status a child is. Instead, the model “know” the probability that a child has to be in each latent status. In the specifically case that you mention, the model can’t tell us in which latent status is a child that have a normal lipid serums level, but it tells us that a child with normal lipids levels has a 70% of probability of being in the latent status one (the no disorder one’s). An ideal model would be one in which the item-response probabilities were all 0 or 1, but in that case the model would be classifying individuals with complete certainty. All models are composed of a component of uncertainty or error (as a regression model for example) and the same happens with LTA models. In their case, this uncertainty or degree of error is reflected in probabilities far from 0 or 1. Otherwise, we would not need a latent model to identify "hidden" patterns, since it would be immediate to classify individuals according to their response to the variables.

After interpreting the probabilities for latent status one, we decided to label it as the no disorders status, since in it all individuals have high probabilities of being in the normal category for all variables. Again, this is the label considered by the researchers, which is susceptible to other interpretation or change. In general, latent status labels must strike a balance between being concise and accurate in describing the pattern of that latent status, since a much more detailed description would be more accurate, but then it would not be a label, but rather a detailed description of the probabilities of that latent status. We provided that completely description in the Results section, so even if we have decided to assign certain labels, the detailed information is available both in the results text and in Table 2.

- Figure 2: risk factors are not labelled and cannot be discriminated.

Response: Based on the reviewer’s comment we have restructured the first and second paragraphs of the Results section, to try to be clearer. The information provided by the reviewer is contained in Table S1 of the supplementary material. It shows the frequency of each disorder among children presenting only one disorder, among those presenting two at the same time, etc... However, the way this table was referenced in the Results section was perhaps inadequate and did not make clear the information it contains. Therefore, it has been reformed.

Despite the high prevalence of hypertension at T1 in the last illustration of the paper.

It is not logical how it is not linked to obesity.

As occurs in adulthood, childhood hypertension (or extremely high blood pressure levels) can be classified as primary or secondary hypertension depending on the aetiology. In this study we refer to primary hypertension (we do not know the specific cause underlying these elevated values) which has a multifactorial origin. One of the main contributing factors in paediatric ages is, to a large extent as noted by the reviewer, the presence of obesity. But also prenatal and postnatal exposures such as tobacco smoke, genetic factors, birth factors such as low birth weight, dietary factors such as high sodium intake, or lifestyle factors, are related to primary hypertension in childhood and adolescence [1]. Some of these factors have been shown to be related to primary hypertension independently of obesity, as evidenced for example by Rosner et al. [2] in the case of high sodium intake. Specifically, it has also been found that the Spanish paediatric population exceeds the recommended sodium intake at ages 9 to 12 years regardless of their nutritional status [3], and that it is positively associated with an elevated diastolic blood pressure between 5 and 16 years old, also regardless of their nutritional status [4]. We find this latent status characterized by hypertension without obesity at 8 years of age very interesting, since it represents those children who have elevated blood pressure values but not related to obesity, and therefore could be related to other independent causes such as high sodium intake, or other of those mentioned above.

We have added a comment regarding this in the discussion of the article (seventh paragraph), which we believe enriches the document, thanks to the reviewer's comment: “Regarding to hypertension, the increasing trend of elevated blood pressure levels in childhood and adolescence in terms of primary hypertension has been mainly related to the also increasing trend of childhood obesity, but also to other factors such as prenatal and postnatal exposures, genetic factors, birth characteristics, dietary habits, or lifestyle factors [1]. Some of them have been shown to be related to primary hypertension independently of obesity, as evidenced for example by Rosner et al. [2] in the case of high sodium intake. Specifically, it has also been found that the Spanish paediatric population exceeds the recommended sodium intake at ages 9 to 12 years regardless of their nutritional status [3], and that it is positive associated with an elevated diastolic blood pressure between 5 and 16 years old, also regardless of their nutritional status [4]. So the latent status characterized only by hypertension is evidenced the group of children with high blood pressure values related to causes independent of obesity, since those who present elevated blood pressure values accompanied by obesity would be classified in the latent status of central obesity.”.

It would be very interesting to know the characteristics of the children classified in this group in relation to these factors (diet, characteristics at birth, pre- and post-natal exposures, etc.) but it is beyond the scope of this study to evaluate the variables that influence whether a child is in one state or another. We are considering this as future research.

References:

1. S. Machado IB, Tofanelli MR, Saldanha da Silva AA, Simões e Silva AC. Factors Associated with Primary Hypertension in Pediatric Patients: An Up-to-Date. Curr Pediatr Rev. Bentham Science Publishers Ltd.; 2021;17:15–37. 

2. Rosner B, Cook NR, Daniels S, Falkner B. Childhood Blood Pressure Trends and Risk Factors for High Blood Pressure: The NHANES experience 1988–2008. Hypertension [Internet]. NIH Public Access; 2013 [cited 2023 Feb 20];62:247. Available from: /pmc/articles/PMC3769135/

3. Partearroyo T, Samaniego-Vaesken M de L, Ruiz E, Aranceta-Bartrina J, Gil Á, González-Gross M, et al. Sodium Intake from Foods Exceeds Recommended Limits in the Spanish Population: The ANIBES Study. Nutr 2019, Vol 11, Page 2451 [Internet]. Multidisciplinary Digital Publishing Institute; 2019 [cited 2023 Feb 11];11:2451. Available from: https://www.mdpi.com/2072-6643/11/10/2451/htm

4. Pérez-Gimeno G, Rupérez AI, Vázquez-Cobela R, Herráiz-Gastesi G, Gil-Campos M, Aguilera CM, et al. Energy dense salty food consumption frequency is associated with diastolic hypertension in Spanish children. Nutrients. MDPI AG; 2020;12.

---

## [Decision Letter · Decision Letter 1]

20 Mar 2023

Cardiovascular risk factors and its patterns of change between 4 and 8 years of age in the INMA-Asturias Cohort

PONE-D-22-28738R1

Dear Dr. Fernández-Somoano,

We’re pleased to inform you that your manuscript has been judged scientifically suitable for publication and will be formally accepted for publication once it meets all outstanding technical requirements.

Kind regards,

Antoine Fakhry AbdelMassih

Academic Editor

PLOS ONE

Additional Editor Comments (optional):

Reviewers' comments:

Reviewer's Responses to Questions

**Comments to the Author**

1. If the authors have adequately addressed your comments raised in a previous round of review and you feel that this manuscript is now acceptable for publication, you may indicate that here to bypass the “Comments to the Author” section, enter your conflict of interest statement in the “Confidential to Editor” section, and submit your "Accept" recommendation.

Reviewer #1: All comments have been addressed

Reviewer #2: All comments have been addressed

2. Is the manuscript technically sound, and do the data support the conclusions?

Reviewer #1: Yes

Reviewer #2: Yes

3. Has the statistical analysis been performed appropriately and rigorously? 

Reviewer #1: Yes

Reviewer #2: Yes

4. Have the authors made all data underlying the findings in their manuscript fully available?

Reviewer #1: Yes

Reviewer #2: Yes

5. Is the manuscript presented in an intelligible fashion and written in standard English?

Reviewer #1: Yes

Reviewer #2: Yes

6. Review Comments to the Author

Reviewer #1: Review comments on Manuscript Number: PONE-D-22-28738. Entitled "Cardiovascular risk factors and its patterns of change between 4 and 8 years of age in the INMA-Asturias Cohort"

I would like to thank the authors for their successful work to address the reviewers' comments. The authors have done great efforts to accomplish this work. They fulfilled all comments and made necessary changes throughput the manuscript. I recommend accepting the manuscript its revised form.

Reviewer #2: (No Response)

7. PLOS authors have the option to publish the peer review history of their article (what does this mean?). If published, this will include your full peer review and any attached files.

Reviewer #1: No

Reviewer #2: No

---

## [Editor Report · Acceptance letter]

3 Apr 2023

PONE-D-22-28738R1 

Cardiovascular risk factors and its patterns of change between 4 and 8 years of age in the INMA-Asturias Cohort 

Dear Dr. Fernández-Somoano:

I'm pleased to inform you that your manuscript has been deemed suitable for publication in PLOS ONE. Congratulations! Your manuscript is now with our production department. 

Kind regards, 

on behalf of

Prof Antoine Fakhry AbdelMassih 

Academic Editor

PLOS ONE